# An Extensible Multimodal Multi-task Object Dataset with Materials

**Trevor Standley**[1]   **Ruohan Gao**[1]   **Dawn Chen**[2]   **Jiajun Wu**[1]   **Silvio Savarese**[1,3]
tstand@cs.stanford.edu

[1]Stanford University   [2]Google Inc.   [3]Salesforce Research

## Abstract

We present EMMa, an **E**xtensible, **M**ultimodal dataset of Amazon product listings that contains rich **Ma**terial annotations. It contains more than 2.8 million objects, each with image(s), listing text, mass, price, product ratings, and position in Amazon's product-category taxonomy. We also design a comprehensive taxonomy of 182 physical materials (e.g., Plastic → Thermoplastic → Acrylic). Objects are annotated with one or more materials from this taxonomy. With the numerous attributes available for each object, we develop a Smart Labeling framework to quickly add new binary labels to all objects with very little manual labeling effort, making the dataset extensible. Each object attribute in our dataset can be included in either the model inputs or outputs, leading to combinatorial possibilities in task configurations. For example, we can train a model to predict the object category from the listing text, or the mass and price from the product listing image. EMMa offers a new benchmark for multi-task learning in computer vision and NLP, and allows practitioners to efficiently add new tasks and object attributes at scale.

## 1 Introduction

Perhaps the biggest problem faced by machine learning practitioners today is that of producing labeled datasets for their specific needs. Manually labeling large amounts of data is time-consuming and costly (Deng et al., 2009; Lin et al., 2014; Kuznetsova et al., 2020). Furthermore, it is often not possible to communicate how numerous ambiguous corner cases should be handled (e.g., is a hole puncher "sharp"?) to the human annotators we typically rely on to produce these labels.

Could we solve this problem with the aid of machine learning? We hypothesized that we could accurately add new properties to every instance in a semi-automated fashion if given a rich dataset with substantial information about every instance.

Consequently, we developed EMMa, a large, object-centric, multimodal, and multi-task dataset. We show that EMMa can be easily extended to contain any number of new object labels using a Smart Labeling technique we developed for large multi-task and multimodal datasets. **Multi-task** datasets contain labels for more than one attribute for each instance, whereas **multimodal** datasets contain data from more than one modality, such as images, text, audio, and tabular data. Derived from Amazon product listings, EMMa contains images, text, and a number of useful attributes, such as materials, mass, price, product category, and product ratings. Each attribute can be used as either a model input or a model output. Models trained on these attributes could be useful to roboticists, recycling facilities, consumers, marketers, retailers, and product developers.

Furthermore, we believe that EMMa will make a great multi-task benchmark for both computer vision and NLP. EMMa has many diverse CV tasks, such as material prediction, mass prediction, and taxonomic classification. Currently, most multi-task learning for NLP is done on corpora in which each input sentence is labeled only for a single task. In contrast, EMMa offers unique tasks, such as predicting both product ratings and product price from the same product listing text.

One important contribution of this work is that EMMa lists each object's material composition. No existing materials dataset has more than a few dozen material types; furthermore, there are no other *large* materials datasets that are object-centric. This prevents models from being able to learn

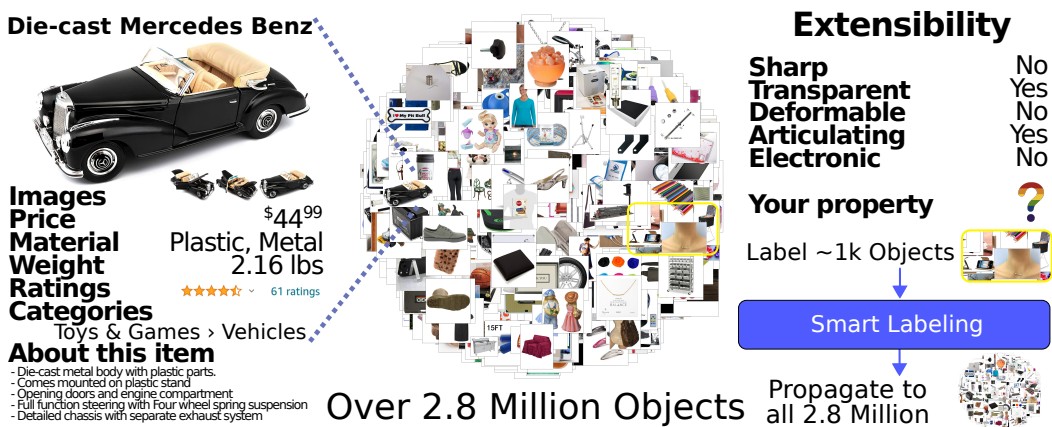

**Figure 1:** We introduce EMMa, an object-centric, multimodal, multi-task dataset of Amazon product listings that contains over 2.8 million objects. Each object in EMMa is accompanied by images, listing text, mass, price, product ratings, position in Amazon's product-category taxonomy, etc. We also introduce a *Smart Labeling* technique that allows practitioners to easily extend the entire dataset with new binary properties of interest (e.g., sharpness, transparency, deformability, etc.) with only hours of labeling effort.

important distinctions about the materials from which an object is made. In contrast, each object in EMMa is annotated with one or more materials from a hand-curated taxonomy of 182 material types (see Figure 2).

Armed with a dataset containing such rich annotations, we developed a technique for adding high-quality binary object properties to EMMa with minimal manual labeling effort, by leveraging all available data for each object. Our technique employs active learning and a powerful object embedding. We envision practitioners adding the labels themselves for their own use cases, obviating the substantial work typically required to obtain and curate high-quality data labels from crowdsourcing services such as Amazon Mechanical Turk.

Our main contributions are threefold. First, we present EMMa, a large-scale, multimodal, multi-task object dataset that contains more than 2.8 million objects. Second, our dataset is labeled in accordance with a hand-curated taxonomy of 182 material types, which is much larger compared with existing material datasets. Third, we propose a Smart Labeling pipeline that allows practitioners to easily add new binary labels of interest to the whole dataset with only hours of labeling effort.

## 2 RELATED WORK

**Multi-Task Learning Datasets**   Many large computer vision multi-task datasets, such as Taskonomy (Zamir et al., 2018), 3D Scene Graph (Armeni et al., 2019), and Omnidata (Eftekhar et al., 2021), offer a large number of tasks for each image. However, unlike EMMa, these datasets are not object-centric and focus on pixel-level prediction tasks. Other multi-task datasets, such as COCO (Lin et al., 2014), NYUv2 (Nathan Silberman & Fergus, 2012), and Cityscapes (Cordts et al., 2016), are relatively small and have only a few tasks, while others are either artificial, such as MultiMNIST (Sabour et al., 2017), or focused on a restricted domain like human faces (Liu et al., 2015).

**Learning from Amazon Data**   Our work is not the first to use Amazon data in a machine learning context. For example, the ABO dataset (Collins et al., 2022) provides listings for 150k objects, about 8k of which have 3D models. Unfortunately, nearly half of the 150k objects are cell phone cases, and the data provided for most objects is in the raw form provided by Amazon. Likewise, image2mass (Standley et al., 2017) curates a dataset of Amazon listings for the purpose of predicting an object's weight given its image, but the processed dataset is only for a single task. The UCSD Amazon review data (Ni et al., 2019) is quite large, containing raw data for 15.5 million products, with a focus on product reviews. We incorporate raw data from the image2mass and UCSD datasets into our dataset, which also contains new data we collected from Amazon. As far as we know, we are the first to take advantage of the bulk of the information in Amazon product listings for machine learning.

**Auto Labeling**  Our work is also related to algorithms for label propagation (Gong et al., 2016; Wang et al., 2013; Luo et al., 2015; Iscen et al., 2019; Douze et al., 2018). While they focus on distances between instances in graphs, we use deep learning predictions from multiple data sources to fill in missing information. There are some similarities here with Tesla's auto-labeling (Elluswamy & Karpathi, 2021), where labels are automatically produced or corrected using other data in the dataset. Other projects, such as Snorkel (Ratner et al., 2017) and Snorkel MeTaL (Ratner et al., 2018), seek to alleviate the data labeling burden by building on top of modeler-created sources of weak supervision. We take inspiration from them in the form of our keyword features (See Section 5).

**Material Classification Datasets**  The largest existing material datasets are OpenSurfaces (Bell et al., 2013), which has  20k images with per-pixel labels from 36 material categories, and the Materials in Context Database (MINC) (Bell et al., 2015), which has 436k images with 23 categories. Other datasets with materials are relatively small but focus on a variety of domains, such as fabric, photography, and objects (Sharan et al., 2014; Gao et al., 2021; 2022; Collins et al., 2022; Bell et al., 2014; Wang et al., 2016; Kampouris et al., 2016; Sumon et al., 2022). In contrast, we have over 2.8 million Amazon product listings with labels from a taxonomy of 182 materials. Moreover, we capture materials that are not necessarily visible on the surface of an object, such as copper inside a wire.

**Attribute datasets**  The MIT-States and UT-Zappos (Isola et al., 2015; Yu & Grauman, 2014; 2017) datasets include object attributes, but are relatively small and focus on crowdsourcing the labels.

## 3 THE EMMA DATASET

EMMa is a large-scale, multi-task, multimodal, dataset of over 2.8 million objects. See Figure 1. First we discuss the data sources and our data cleaning process. Then, we present the diverse data attributes accompanying each object in our dataset. Finally, we summarize some statistics of our dataset.

### 3.1 DATA SOURCES

We collected data from three sources. We started with the Amazon Selling Partner API (ASPA), from which we collected about 2 million items. Because of the QPS limit, this took about three months. We augmented this data with other publicly available datasets. We used the raw image2mass dataset, which has about 3 million listings, and the UCSD product rating dataset, which has about 15 million listings. In all three data sources, each entry is keyed by the Amazon Standard Identification Number (ASIN). Unfortunately, each data source was incomplete. For example, the UCSD data often lacked weight, materials, and size information; the ASPA data lacked product descriptions, features, categories and ratings; and the image2mass data lacked descriptions, categories, and ratings. Moreover, each entry from the image2mass and ASPA sources had only a single image. To overcome this limitation, we join these datasets. This ensures that each entry has as all the available information.

This resulted in approximately 18 million entries, which were aggressively filtered through several steps (detailed in the Appendix). The entire filtering process resulted in a dataset with 2,883,698 instances. This was partitioned into 2,806,806 training instances, 26,535 validation instances, and 26,941 test instances, and 23,416 calibration instances. The test and validation instances were chosen from the set of instances that have all attributes filled-in and have more than one image and more than one review. Furthermore, more aggressive duplicate-detection was applied to these sets to ensure they don't contain objects that are similar to objects in the training set. Product listings from the image2mass Amazon test set were excluded from the training set so that we can compare with their results below.

### 3.2 DATA ATTRIBUTES

EMMa contains the following attributes:

**Images**—Every product listing contains one or more product images. There are 7,389,213 images total, an average of 2.56 images per listing (though the distribution is skewed). We use the original resolution (500x500) in our models. In this work, we never treat images as an output; however, we

think that the dataset could be useful for image generation. We also provide a representation of the images in each listing by taking the vector sum of the ConvNeXT-Large(Liu et al., 2022) features for all images in the listing.

**Text**—Every product listing contains a title. Titles can be surprisingly descriptive, often stating materials, object categories, or uses. 88% of our listings contain one or more bullet points, i.e., Product Features. Most of the textual information about a product is contained in these features. 63% of our listings contain a description. Descriptions can describe the product, the brand, or even repeat information already contained in the features. Together, the title, features, and description are concatenated to create the input for our text models. We provide a representation from an all-mpnet-base-v2 (HuggingFace, 2021) of this text as part of the dataset. We think that this dataset could also be useful for captioning by using the text as an output, although we don't explore this in the present study.

**Weight (Mass)**—77% of our final dataset contains the product's weight.

**Price**—69% of our final dataset contains price information. Prices are missing for some products because some listings remain on the website even though the products are no longer available for purchase, and the price will be unavailable in such cases. When multiple data sources contained different prices for the same product, the median price was used.

**Materials**—30% of our product listings contain materials. The raw data for materials is simply a list of seller-entered strings. Sellers can put anything into the materials entry, and so it had to be heavily cleaned and processed. We developed a taxonomy of 182 material types as shown in Figure 2. We used heuristics to match seller-entered materials strings to nodes in the hierarchy when possible in order to create the final materials labels. Items can be made out of multiple materials. Very often, it is not known which leaf node an object has, but rather only an internal node. For example, we may know that an object is made of plastic, but not which kind of plastic. Special attention was given to the quality of the materials label. 11% percent of the entries in the calibration set were manually labeled, focusing on the most difficult instances. We use this calibration set to help train and calibrate the probabilities of our materials models. Furthermore, we applied the smart-labeling process (Section 5) to 19 of the material nodes (nylon plastic, nylon fabric, cardboard, cardstock, diamond, liquid, electronics, food, gold, genuine leather, lace, marble, faux-fur, cherry, leather, fabric, wood, metal, and plastic) to refine the the labels for every object in the dataset.

**Category**—70% of our final dataset contains a product category, which specifies the position of a product in Amazon's product taxonomy, e.g., Home & Kitchen → Kitchen & Dining → Cookware → Pots & Pans → Woks & Stir-Fry Pans. The hierarchy implied by the data contains 11,592 nodes at a maximum depth of 6. Even though Product Type and Category are both semantic classifications of the objects, they have surprisingly little overlap. Preliminary experiments show that models trained to predict one don't benefit much from the inclusion of the other as an input.

**Size information**—Most of the final dataset includes some form of size information, such as item dimensions or package dimensions. We used this data as an input for training models to predict mass as well as models to fill in missing data. However, we do not predict size information as a task.

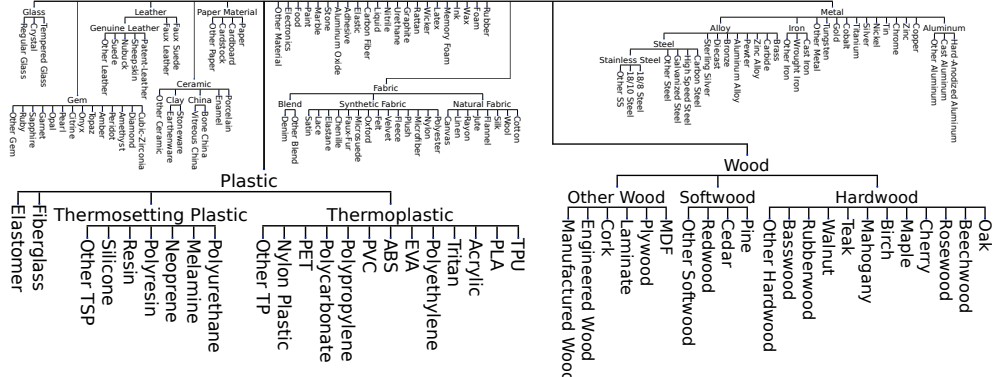

**Figure 2:** Our materials taxonomy. Nodes were chosen to capture most of what Amazon sellers were writing in the materials field. The plastic and wood branches are enlarged for legibility. See the Appendix for our materials taxonomy design philosophy.

**Reviews and ratings**—75% of the listings have at least one extracted review. Each review comes with a product rating. We predict these ratings as a task and use all-mpnet-base-v2 to extract features from reviews to use as input.

## 3.3 DATASET STATISTICS

Figure 3 shows the diversity of the dataset. The price distribution shows a good spread of both inexpensive and expensive items. About 9% of objects in EMMa cost more than $200. The objects also have a wide variety of masses, with many objects weighing less than a tenth of a pound and many weighing more than 100 pounds. Moreover, the dataset contains objects with a diverse set of materials. 78 nodes in the materials taxonomy have more than 10k instances labeled with them, and 91% of the nodes have more than 500 instances labeled with them. On average, each object has 4.9 material labels. Furthermore, 1,049 category nodes have more than 1,000 listings. This subset of the data could be used to compare with ImageNet 1k. Finally, it's rare for a category to be dominated by a subcategory. On average, a node's largest child contains only 53% of the product listings within a node.

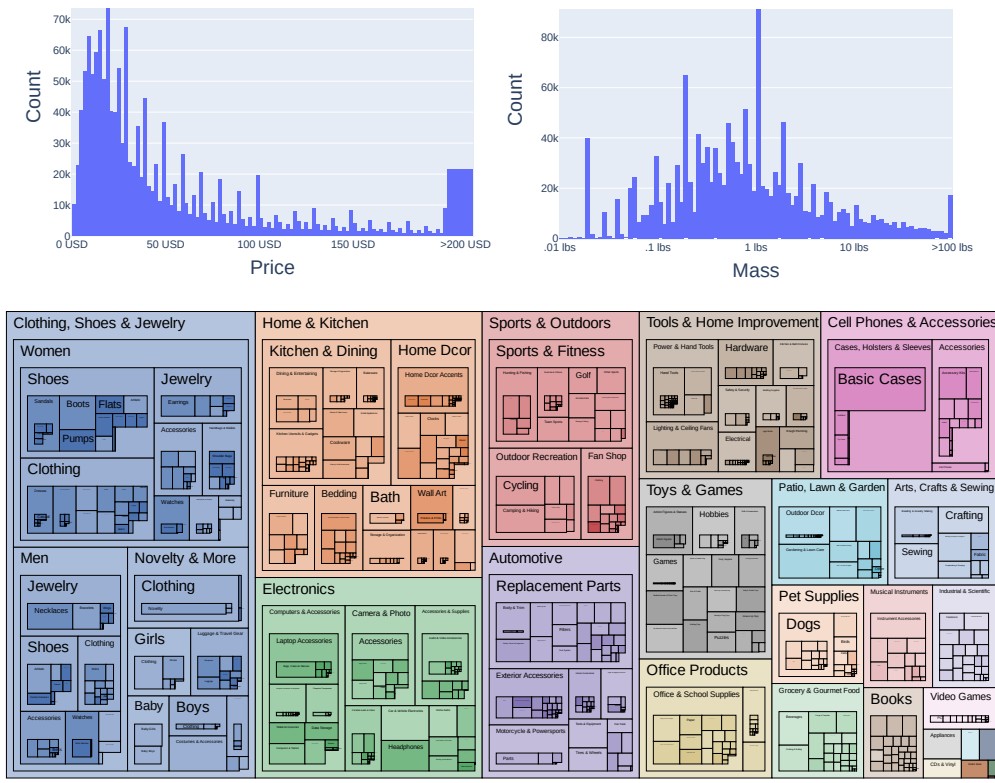

**Figure 3:** (Top left) A histogram of EMMa's prices. (Top right) A histogram of EMMa's masses (log scale). (Bottom) A treemap showing the diversity of the categories in EMMa.

## 4 MODELS AND SYNTHETIC LABELS

Although our final dataset contains all the attributes mentioned in Section 3.2, the raw source data did not have every attribute for every object. For example, only about 875k of the listings had material information. This missing information complicates multi-task learning and reduces the amount of available supervision. However, missing attributes can often be accurately inferred from the available attributes, especially the product listing text. For each of the possible missing attributes except for size (price, mass, materials, categories, and ratings distribution), we trained a model to predict that attribute given all other available attributes. We use these models to fill in the missing information.

| | Price (MnRE) ↑ | Mass (MnRE) ↑ | Materials (F1) ↑ | Category (Acc) ↑ | Ratings (KL-d) ↓ |
|---|---|---|---|---|---|
| SINGLE IMAGE GEN 1 | 0.649 | 0.640 | 60.9 | 82.6 | 0.155 |
| SINGLE IMAGE GEN 2 | 0.653 | 0.649 | 72.7 | 84.5 | 0.153 |
| TEXT GEN 1 | 0.710 | 0.693 | 79.8 | 91.6 | 0.148 |
| TEXT GEN 2 | 0.704 | 0.695 | 85.3 | 91.9 | 0.147 |
| EVERYTHING GEN 1 | 0.736 | 0.773 | 80.0 | 92.1 | 0.138 |
| EVERYTHING GEN 2 | 0.743 | 0.788 | 86.0 | 91.9 | 0.136 |

**Table 1:** The performance of our multi-task text and image models compared with the performance of the models that take all available data. MnRE (Standley et al., 2017), F1, Acc, KL-d denote the minimum ratio error, F1 score, accuracy, KL-Divergence, respectively. ↓ lower better, ↑ higher better. Since we use the Everything models to fill in missing data, the performance we see here on the test set is the accuracy we should expect for the synthetic labels. See the Appendix for a discussion about metrics and losses.

These models were trained using a late-fusion strategy. We fine-tuned a computer vision model (ConvNeXT base pretrained on ImageNet 22k (Deng et al., 2009)) to simultaneously predict price, mass, materials, categories, and ratings from the product images. Likewise, we fine-tuned an Transformer model (all-mpnet-base-v2) to predict the same attributes from the product listing text. We then extracted and standardized feature embeddings from these two models (both are vectors of length 768). Finally, we trained task-specific models that take in these standardized embeddings as well as all the other available attributes. Early experiments showed that this strategy resulted in little to no loss in prediction quality compared to end-to-end training while being much more tractable. Moreover, this strategy produced useful embeddings and allowed faster model iteration. Each task has a specific decoder architecture that was manually designed. See the code for details. See the Appendix for technical details about model training and task losses.

Assuming that the predicted labels are at least somewhat accurate, we can use them to retrain the models and make even better predictions. In turn, with better predictions we can train better models. This is a form of expectation-maximization (Dempster et al., 1977). We refer to the models trained on the original data as generation 1, and the models trained on the dataset with missing attributes filled in as generation 2. Preliminary experiments showed diminishing returns after generation 2.

Table 1 shows the performance of all of our models on the test set. As expected, models that take images, text, and the other attributes into account do better than the models that take just a single image or just the product listing text. Furthermore, we see a general improvement for most tasks when working with filled in data (Gen 2) on all three types of models. Finally, we note that the text models do a surprisingly good job capturing the relevant information. This is particularly surprising in the case of mass, which is a physical property that is usually not present in the listing text.

We use these models to produce synthetic labels to fill in the missing data. For the specific case of the materials task, the Amazon listing may only specify that an object is made out of a mid-level material in the taxonomy (e.g., metal). These models can predict a probability distribution over the types of metal, i.e., steel [0.4], aluminum [0.6], etc. This allows us to fill in probabilistic estimates for the types of metal, leading to a training label that is a full distribution over the 182 materials in our hierarchy. We use an iterative algorithm to enforce that these probabilities make sense, i.e., the probability of a parent must be between that of its highest child and the sum of its children's probabilities. The parent's probability need not be *equal* to the sum of its children's probabilities because a product can contain more than one material subtype (e.g., a chess board can be made of both cedar and pine). Our iterative algorithm also fixes inconsistencies between smart-rated material labels and other labels.

For mass prediction, we compare our image model to the the model from (Standley et al., 2017) in Table 2. Our model outperforms theirs on their publicly available test set despite lacking a "geometry module," though we are using a more recent backbone network (ConvNeXT-Base vs 2x Xception

| | Our Amazon Test Set | IMAGE2MASS Amazon Test Set |
|---|---|---|
| IMAGE2MASS | – | 0.672 |
| OURS | 0.649 | **0.709** |

**Table 2:** The performance of our image model for mass predictions compared with the model from IMAGE2MASS (Standley et al., 2017).

(Chollet, 2016)) and we have a larger dataset. Our model doesn't need to be isolated on a pure white background and is predicting all of our other tasks in addition to mass. According to the study in the image2mass paper, this model performs better than humans.

Filling in missing information not only helps to produce better models, but also improves the object embedding we train in Section 5. If an attribute such as mass were to be missing in a bottleneck model, we would have to input a fixed value (such as zero) and the model wouldn't be able to distinguish that from an object with that fixed mass.

## 5 Extensibility, Smart Labeling, and Object embedding

One of the major problems in modern machine learning is collecting sufficient high-quality data to train a desired model. A common process in academia involves the use of a crowdsourcing platform such as Amazon Mechanical Turk. This is costly and comes with a whole host of other problems, not least of which is the difficulty of communicating exactly how one's data should be labeled. In order to overcome both the cost and communication problems, we outline our *Smart Labeling* process. This process leverages active learning as well as two key properties of our dataset, the ability to use all data about each object to create a powerful but compact object embedding, and the text itself.

**Object Embedding**   Leveraging all known and filled-in attributes for every object, we developed a powerful object embedding. This embedding is a vector of length 2048 that captures most of what we know about a product listing and is trained using a simple network with a single hidden bottleneck layer. The input is a standardized vector containing text and vision embeddings in addition to mass, price, materials, categories, and ratings (Figure 7 illustrates our architecture). The output was trained to match the input using L2 distance and achieves an average standardized L2 of 0.0054, which implies that each input feature is reconstructed to within 7.3% of its variance on average. More details in the Appendix.

**Smart Labeling Framework**   We built a Smart Labeling tool (see Figure 6 for a screenshot) and developed a process for quickly generating high-quality and high-impact training data for adding a new object property to our dataset. The goal of the framework is to train an accurate linear logistic regression model. Predictions from that model can be used to label the rest of the dataset.

The discriminatory power of any linear model is limited by the input features. This is why it is so important to train on top of a powerful object embedding like the one we developed. But as powerful as the embedding is, any fixed set of features can be a limitation. Therefore, in addition to the embedding, we support using any number of custom, task-specific features that can be added to increase discriminatory power as input. These task-specific features can be added using the Smart Labeling tool and can simply be derived from keywords related to the desired property (see below for details). The Smart Labeling tool can be used in one of several modes: **features** (for adding task-specific features), **word-search** (for seeding the training set with a small number of positive and negative examples), **active-learning** (for fine-tuning the decision boundary), **correction** (for finding and correcting errors in the model), or **review** (for finding and correcting labeling errors made in the previous steps). A flowchart of this process is presented as Figure 4.

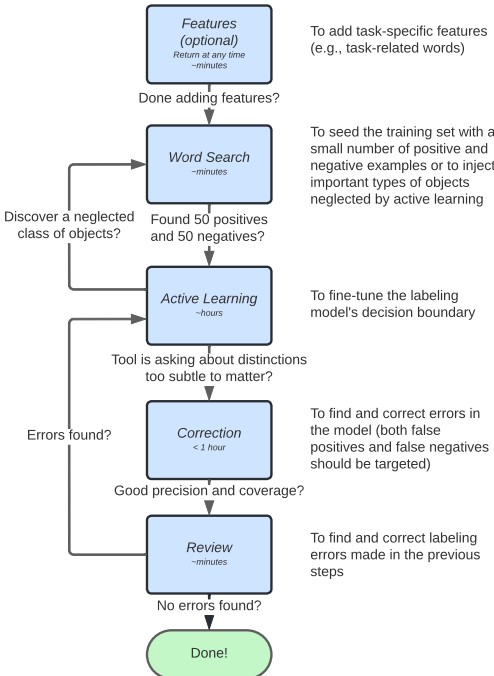

**Figure 4:** A flowchart showing the progression of the Smart Labeling process. This chart should be treated as a guideline and need not be followed exactly.

As an example, let's say that we want to add the **is_container** property to the dataset. First, we use **features** mode to add as many container-like words as possible, which we use as binary *text-matching* features. Each feature is positive iff the listing text contains the corresponding word. They can be added or modified at any time during the Smart Labeling process. They are not necessary, but it often helps to start with some. We'll add *cup*, *mug*, *box*, *container*, *holds*, *basket*, *leak*, *bottle*, *jar*, *bag*, *flask*. Conceptually, these can be regular expressions, or even arbitrary functions that take an object as input and produce a real-valued output somewhat related to the property (like in Snorkel), but we've only tried simple text matching strings.

Then, we use **word search** mode to create the initial training set. We can start with the word "basket". The system will search through listing texts for the word "basket" and show matching product images to the user in a random order. The user can quickly mark all objects that are actually containers as positive instances (left-click) and all the clearly non-container objects as negative instances (right-click). The user can also middle-click on any product image be taken to the Amazon product listing page. In this phase, it's best to skip any object for which it is not easy to determine the label. More matching products can be brought up by clicking to the next page. New search terms can be entered until the initial training set is large enough. Ideally, at least 50 positives and 50 negatives are collected, but we don't need more examples because active learning mode is more efficient for improving accuracy. Nevertheless, we can return to word search mode at any time if, for example, we notice that the model is failing to surface a certain class of objects for labeling.

Next, we move on to **active learning** mode, for which we use uncertainty sampling (Lewis & Gale, 1994). A logistic regression model is trained on the existing labeled data, and predictions are obtained for a sample of unlabeled candidate objects. The objects with predicted probabilities closest to $0.5$ are then presented to the user. The user can then label any or all of these and advance to the next page, at which point the logistic regression model will be retrained and a new candidate set will be chosen and presented to the user. After labeling several pages, it's common to start encountering corner cases, such as, "is a phone case a container?" It helps to record the decision to ensure consistency. This phase tends to take the most time per object because the system quickly hones in on the difficult cases. When most objects presented by the tool either cannot or need not be labeled (because their labels don't matter for the use case), it's time to move on to correction mode.

**Correction** mode can be used to improve either precision or recall. This mode works just like active learning mode, except that instead of presenting the user with objects that have probability near $0.5$, it presents users with objects that have currently predicted probabilities within a user-specifiable range. If the range is $0.5 - 1.0$, then the system will present the user with objects that are currently classified as positives. In this case, the user should focus on correcting false positives and can ignore correct or difficult items. Correcting even a few false positives can have a large positive impact model performance. On the other hand, setting the probability range to $0.0 - 0.5$ allows a user to correct false negatives. In practice, many of the properties a user might want to add to the dataset are rare (Bommannavar et al., 2014). In this case, it might be more useful to set the range to be something like $0.1 - 0.5$, because otherwise the false negatives might be difficult to find.

Finally, **review** mode lets the user review the data they have already labeled. In review mode, 20-fold cross-validation is used to find how likely each example is to be mislabeled. Instances are presented to the user in order of descending score. This allows a user to quickly find a (hopefully) small number of labeling errors that they may have made and correct them. Since we are using active learning, labeling errors have a large effect on prediction quality. Users should return to active learning mode if they find labeling errors.

When precision and positive rate are sufficiently high for the intended use case, we can use the model to predict a probabilistic label for every object in the dataset.

**Smart Labeling Evaluation**  We added five custom properties to the dataset using Smart Labeling: Sharp, Transparent, Deformable, Articulating, and Electronic. See the Appendix for property definitions, manual labeling criteria, and the text-matching features we used. See Figure 5 for some qualitative examples of products labeled by our models. We also created a test set for each class. Unlike the training set, the test set was sampled uniformly from the set of all products. Sample products that could not be manually labeled with confidence were discarded. Table 3 shows performance metrics for each property. F1 and accuracy scores in the 90s were achieved on all five

|  | # Train | # Train + | # Test | # Test + | # Errors | Precision | Recall | F1 | Acc |
|---|---|---|---|---|---|---|---|---|---|
| SHARP | 1331 | 633 | 1504 | 25 | 4 | 92.0 | 92.0 | 92.0 | 99.7 |
| TRANSPARENT | 2217 | 877 | 706 | 46 | 6 | 93.5 | 93.5 | 93.5 | 99.1 |
| DEFORMABLE | 636 | 350 | 253 | 150 | 9 | 96.7 | 97.3 | 97.0 | 96.4 |
| ARTICULATING | 1399 | 817 | 403 | 148 | 18 | 93.3 | 94.6 | 94.0 | 95.5 |
| ELECTRONIC | 525 | 248 | 301 | 63 | 5 | 96.8 | 95.2 | 96.0 | 98.3 |

**Table 3:** Performance of using the Smart Labeling technique to add five properties to EMMa. The sizes of the test and training sets are shown, as well as the number of positive examples in each set (# Train +, etc.). The numbers of errors, precision, recall F1, and accuracy are shown.

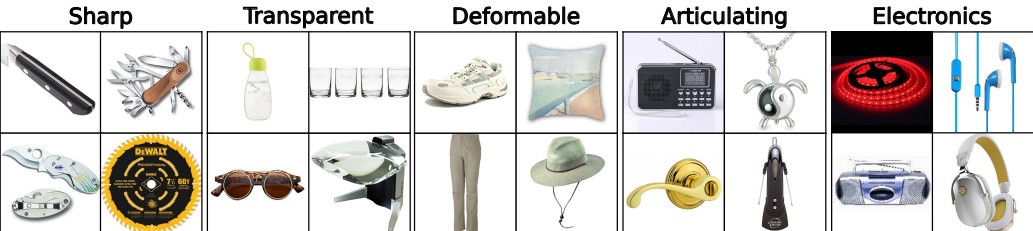

**Figure 5:** Some qualitative examples of products with highly positive inferred labels. More examples can be found in the Appendix, along with examples of base dataset attributes, such as heavy and expensive.

tasks. We see that rare properties can also be easily annotated. The uncertainty sampling approach helps to balance the number of positive and negative examples given to the model.

**Smart Labeling Advantages** By labeling their own data, a practitioner overcomes communication friction, can leverage their own domain-specific expertise, and can tailor the labels to their specific use case. For example, a roboticist could annotate which objects their particular robot could likely pick up. As another example, whether certain objects should be considered "sharp" depends on the specific use case: a hole puncher may be considered sharp if deciding which objects can cut a string and not sharp if deciding which objects can be wielded safely by a robot. Often, these corner cases are hard to predict a priori and are best handled when they appear in the labeling process. Such corner cases are likely to appear in the Smart Labeling process due to the use of uncertainty sampling.

Another advantage of the Smart Labeling technique is that the predicted probabilistic labels are more powerful than standard binary labels. Annotation errors are likely to be less certain, which means that any model learning from these annotations won't devote as many resources to match them.

Our framework for creating new labels efficiently can be useful in other situations as well. For example, we used a similar process with image embeddings to detect and filter out unhelpful product listing images, such as those of the packaging or of warranty labels.

In order to allow EMMa to expand and improve, we will host not only the core dataset, including our manually added properties, but also any properties that the community develops for EMMa and wants to share. This way, both the number of object properties and the ease of adding new properties will grow over time and EMMa will become increasingly useful to machine learning practitioners.

# 6 CONCLUSION

EMMa is an object-centric, multimodal, and multi-task dataset of over 2.8 million Amazon product listings with rich annotations. EMMa's objects are annotated with material labels from a comprehensive taxonomy of 182 material types. We also propose a Smart Labeling framework that allows practitioners to accurately add new binary properties with only hours of effort.

## ETHICS STATEMENT

We are committed to addressing potential ethical concerns in our dataset. We note that Amazon actively enforces its community standards on product listings and reviews. We rely on this to ensure

our dataset does not contain overt racism, foul language, nudity, illegal products, and harassment. We also employ Smart Labeling to mark images with human models or products of a sexual or religious nature, so they can be filtered when necessary. In addition, smart-labeling empowers practitioners to clean and re-balance the dataset for their own ethical purposes. For example, practitioners can use Smart Labeling to annotate the gender of human models and then use stratified sampling to re-balance the distributions during training to ensure that no gender is underrepresented.

ACKNOWLEDGMENTS

The work is in part supported by ONR MURI N00014-22-1-2740, NSF CCRI #2120095, NSF RI #2211258, AFOSR YIP FA9550-23-1-0127, the Stanford Institute for Human-Centered AI (HAI), Adobe, Amazon, and Meta.

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

# A    APPENDIX

**Filtering and Data Cleaning**    After merging based on ASIN, we had approximately 18 million objects. However, the data still contained many duplicates, consisting of the same product being sold under different listings. This was especially true for the UCSD data. When duplicates or near-duplicates were detected, we only kept the entry with the most attributes filled in.

Duplicate detection was accomplished using the pretrained NLP embeddings and the pretrained ImageNet embeddings. Any two entries for which either of the two embedding types were too close, were removed.

Next, we discarded any entries without titles or without images. Furthermore, any entry lacking mass, materials, and category was discarded (if one was present, the entry was kept).

Then, we built a model to detect entries for which the listing text did not match the images and removed those entries. Finally, some entries had only generic images that don't show the product and other entries had no physical form, such as downloadable software. We used an early version of Smart Labeling (Section 5) to find and discard such problematic entries.

Unfortunately, the data sources were collected at different times (from 2017 to 2021) and we found that a seller will occasionally replace the product listing information for one product with the information of another they hope to sell, perhaps piggybacking on the positive reviews of the original product. We used heuristics to detect this and ensure that merged entries with the same ASIN were indeed for the same product. Otherwise the entry was discarded.

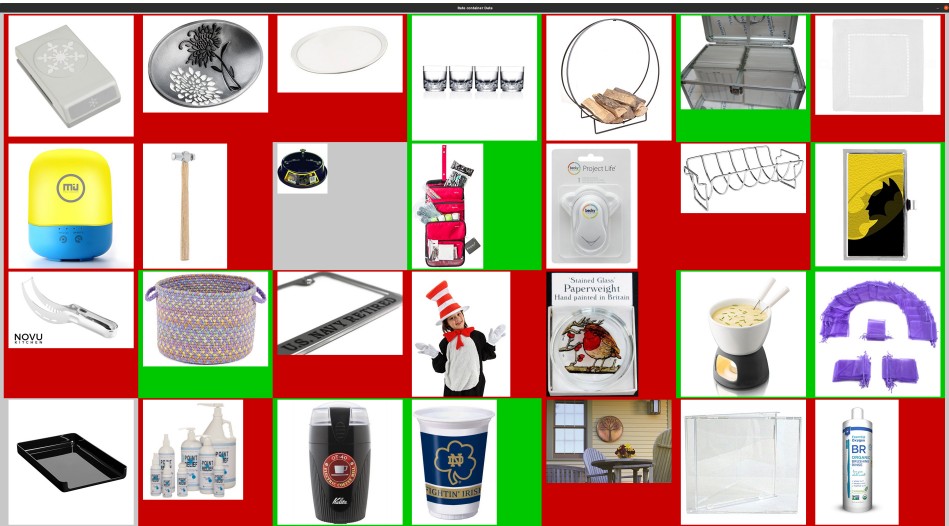

**Figure 6:** A screenshot of our Smart Labeling tool's UI during the lableing of a hypothetical class, is container. Red entries are manually marked not in the class. Green entries are manually marked in the class. Unmarked entries are yet to be labeled or can remain unlabeled.

**Labeling Criteria**  The following are the labeling criteria we used for the Smart Labeled properties from section 5. We tried to include numerous categories with information that isn't obviously contained within the Amazon data.

Sharp—Objects are considered sharp when they are used to cut or poke holes into things. Objects with internal sharp parts, such as craft hole punchers are not considered sharp. We used the following keyword features, "needle, knife, blade, razer, scissors, pin, sharp, dull, knitting, serrated, saw, sharpener" for the sharp labeling process.

Transparent—Objects are considered transparent if they can be seen through to the other side or to objects placed inside of them. For example, a typical watch is not considered transparent because you can only see the watch face through the glass, but a picture frame is considered transparent because you can put a picture inside it. Translucent objects are not considered transparent. As is often the case for such classes, there's a continuous gradient from transparent to translucent where we had to use our own judgement. In general, details must be visible through the object for it to be considered transparent. We used the following keyword features, "needle, knife, blade, razer, scissors, pin, sharp, dull, knitting, serrated, saw, sharpener" for the transparent labeling process.

Deformable—Objects are considered deformable if typical use-case force can change their shape. Fabrics are considered deformable, as are molding clays, gels, bendable wires/springs etc. We used the following keyword features, "shirt, cloth, fabric, clay, bendy, silicone, rigid, hard, stiff, cotton, polyester, flex, paste, soft" for the deformable labeling process.

Articulating—Articulating objects are rigid bodies connected in a movable way. Things with hinges or doors are articulating. As are connected wheels or other items that have parts that rotate. Chains and linked jewelry is also considered articulating. We used the following keyword features, "glasses, box, hinge, door, chain, telescoping, adjustable, book, fold, folding, folio, twist" for the articulating labeling process

Electronic—Anything with electronic components or wiring is considered electronic. Lamps are considered electronic, so are cables that send signals or power. Batteries or anything with a battery is considered electronic. So is anything with an AC plug. Automatic camera lenses and accessories that use power are also considered electronic. Finally Auto-parts are considered electronic if they use electricity, however, these were very difficult to rate with accuracy because we do not have auto-parts expertise, so many of these were left blank and not considered in either the train or test sets. We used the following keyword features, "power, battery, electr, automatic, smart, case, led, usb, tool, mechanical, windup, charge, automatic, quartz, 0w, watt" for the electronics labeling process.

Packaging is not considered part of the object, and is not considered for labeling. Generally, an object is considered positive for a property if any part of an object has the property. However, sometimes when an object overwhelmingly does not have a property, but has it on a small part it is not. For example, when shoes/clothes have zippers we still do not consider them articulating.

**Losses:** For the price and mass tasks, we use ALDE as in (Standley et al., 2017). For materials, we use the mean of the binary cross-entropy loss over all 182 material types. When a data item is labeled with an internal node in the materials taxonomy, the descendants of that node are excluded from the mean calculation so that models aren't penalized for being more specific than the label. For example, if an object is labeled as plastic, predictions of thermoplastic or acrylic are not penalized. For categories, we use a cross entropy loss at each level of the taxonomy. We take a weighted mean of the loss at every level of the taxonomy. In other words, it's more important for the prediction to match at higher levels in the taxonomy. Furthermore, we only consider predictions made for the correct parent at any given level. In other words, if an object is in the Home & Kitchen $\rightarrow$ Cookware $\rightarrow$ All Pans category, we do not consider the model predictions for Automotive $\rightarrow$ Replacement Parts at level 2 and we do not consider the predictions for Home & Kitchen $\rightarrow$ Bath $\rightarrow$ Towels for level 3. For ratings, we use KL-divergence and weight each product listing by the log of the number of reviews it has.

**Metrics:** For the Price and Mass tasks, we report the Min Ratio Error (MnRE) as defined in (Standley et al., 2017). MnRE= $\min(\frac{p}{t}, \frac{t}{p})$ where $p$ is the prediction and $t$ is the ground truth. This metric is nice because perfect predictions result in a score of 1.0 and the worst possible predictions have a score of 0.0. For materials, we report the average F1 score of each object, where the F1 score of an object is the average F1 score of each material classifier. For categories, we report the accuracy of each predicted class averaged first over level in the hierarchy and then over instances. For Ratings, we simply report the loss (KL-Divergence), though we note that the Pearson's $R^2$ between the average rating and the predicted average for an object also improves during training.

**Training Details** All models were trained using Pytorch and the AdamW optimizer. All models used a batch size of 512. All gradients were clipped to 32. Task losses were combined using a weighted average. The weights are in Table 4. Text and vision models use an encoder-decoder architecture. For both, the encoder is frozen during the first epoch. Text and vision models were trained on 8xTitan RTX. Each took about three days to train. Everything models were trained on a 1xTitan RTX wokstation in about 12 hours. Each model type has its own learning rate schedule and weight-decay, which are listed in Table 5.

| Price | Mass | Materials | Categories | Ratings |
|-------|------|-----------|------------|---------|
| 5.0 | 5.0 | 140.0 | 1.0 | 1.0 |

**Table 4:** The loss weights used for Text and Vision models. This should not be confused with task importance because losses have different scales. Weights were initally chosen using GradNorm (Chen et al., 2017) and then hand tweaked from there.

**Other Useful Data Fields** In addition to the fields described in Section 3.2, we extract the following fields:

**Product Type**—Every product listing contains a Product Type designation because it is required to start a listing. The product type is used by the seller UI to determine what information to ask about.

|  | TextToAll | VisionToAll | AllToPrice | AllToMass | AllToMaterials | AllToCategories | AllToRatings |
|---|---|---|---|---|---|---|---|
| Weight Decay | 0.01 | 0.001 | 0.1 | 0.1 | 0.03 | 0.01 | 0.02 |
| Initial Learning Rate | 0.001 | 0.002 | 0.002 | 0.002 | 0.02 | 0.02 | 0.0002 |
| Epochs | 50 | 50 | 100 | 100 | 100 | 100 | 100 |

**Table 5:** Manually tuned hyperparameters for our models. Initial learning rates were reduced by a factor of 100 during the course of training.

Examples include PILLOW and WIRELESS_ACCESSORY. There are 1,296 unique classes, but their counts are heavily skewed. Product Type does not seem to be exposed in the buyer user interface.

**Package Weight**—85% of our final dataset contains the package weight, which includes not only the weight of the product, but also the weight of the packaging. We feed this information to the models that we use to fill in missing attributes. It's particularly informative for filling in a product's weight. Sometimes the Package Weight listed is less than the listed Weight, which indicates an issue with one or both labels.

**Object Embedding** Figure 7 shows the architecture for creating our object embedding. The input is a 4,542-dimensional vector composed of the image and text embeddings mentioned in Section 4, the original pre-trained image and text embeddings mentioned in Section 3, as well as price, mass, materials, ratings, a 128-bit embedding for the product type, and a 128-bit embedding for the product category. Our object embedding is the standardized activation of the bottleneck layer. We use an L2 loss to train our network. We use a batch size of 512 and a learning rate of 3e-3 to 3e-7, approximately halving every 4 epochs. We train for 50 epochs. We achieve a final loss of 0.0054, which implies that each input feature is reconstructed to within 7.3% of its variance on average.

Using this embedding, we can reconstruct the category with 99.8% accuracy, the materials to an F1 of 98.45. The mass to a Pearson's $R^2$ of 0.998 correlation with the ground truth, the price to a Pearson's $R^2$ of 0.960.

**Synthetic label Sanity Check** In order to sanity check our synthetic labels, we randomly chose 100 products from the training set that initially lacked category labels. We examined the synthetic labels and found that 82 of those labels were either perfect or very good descriptions of the product. We repeated this experiment with a random sample of objects that included categories from Amazon. We found that 86 of those had good descriptions.

**Materials Taxonomy Design Choices** The taxonomy was designed in a data-driven way. Many entries in the raw datasets might not be considered materials. For example, 'plush' was present in thousands of products' materials entries. We decided to include materials like these for a number of reasons. First, many of these imply their parents, i.e., microfiber implies fabric.

Second, philosophically, most materials we think about are not pure elements, compounds or mixtures. Leather and wood, for example, are about the configuration of molecules, not the base molecules themselves.

Finally, we can envision uses for knowing about these types of "materials". For example the weave of fabric. Some weaves are denser or more expensive than others, so that has implications for price and mass, and may even be a useful signal for custom features.

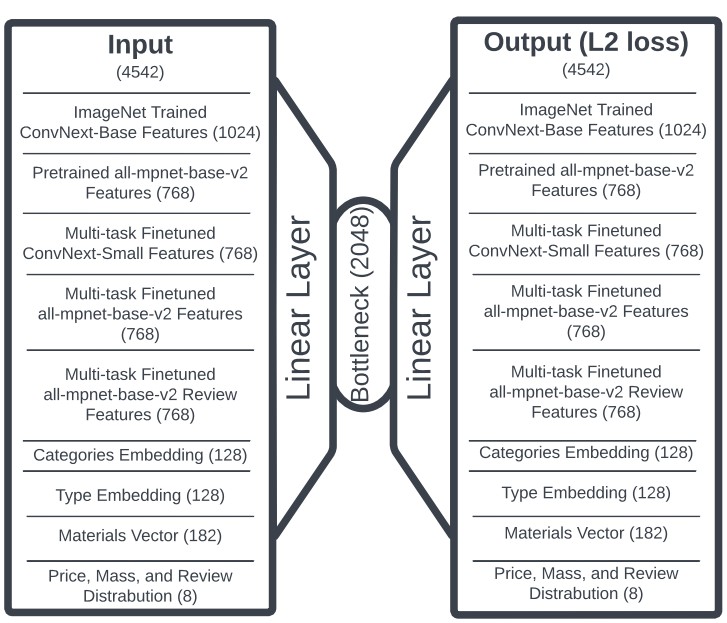

**Figure 7:** Our architecture for creating the powerful object embedding.

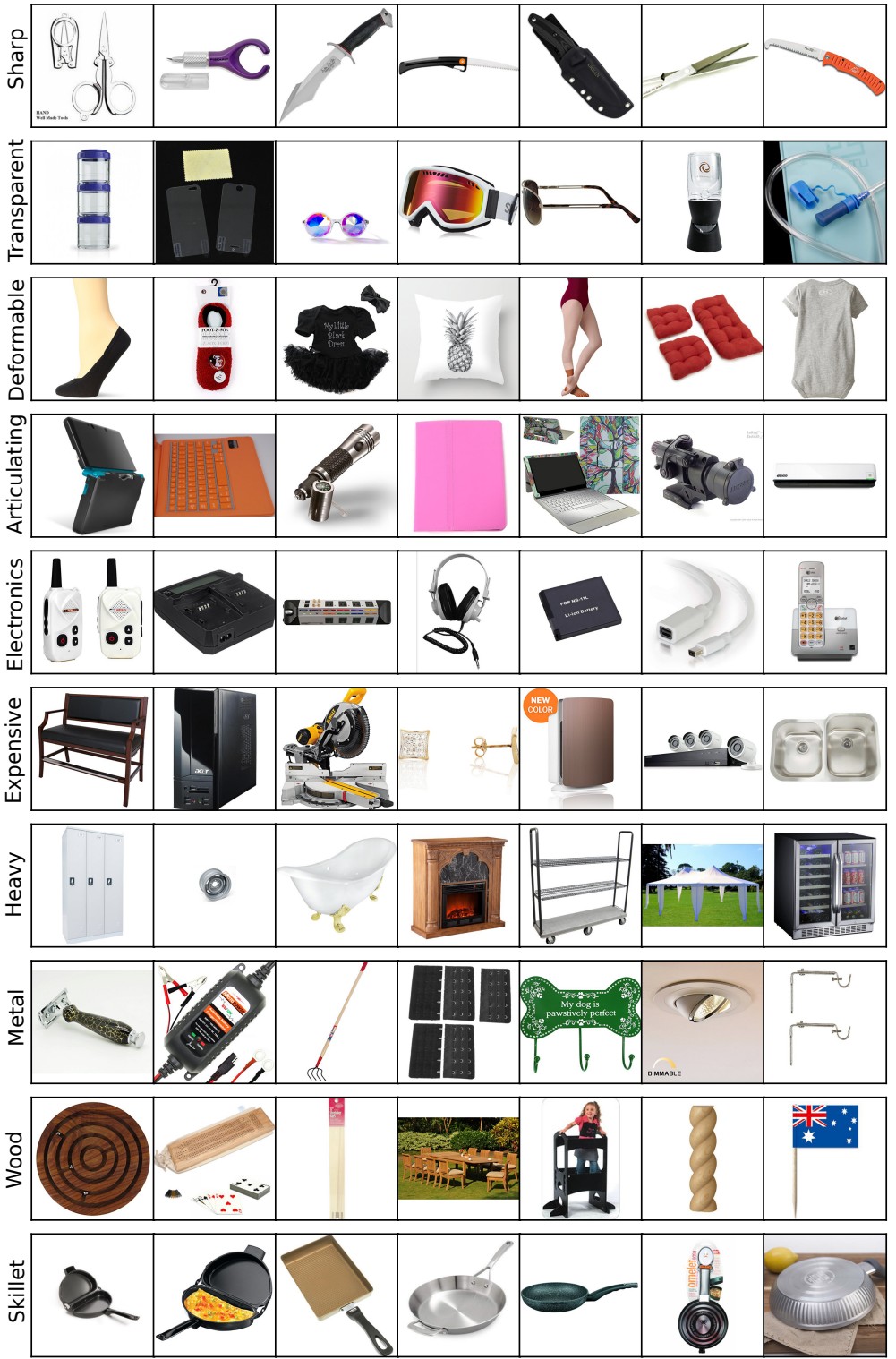

**Figure 8:** A qualitative look at our dataset. The top 5 rows have high probabilities of having the labeled custom property. Expensive objects are those that cost more than 300 USD. Heavy items are more than 100 pounds. Metal and Wood are material attributes. Finally, we see a selection from the Home & Kitchen → Cookware → All Pans → Skillets category.

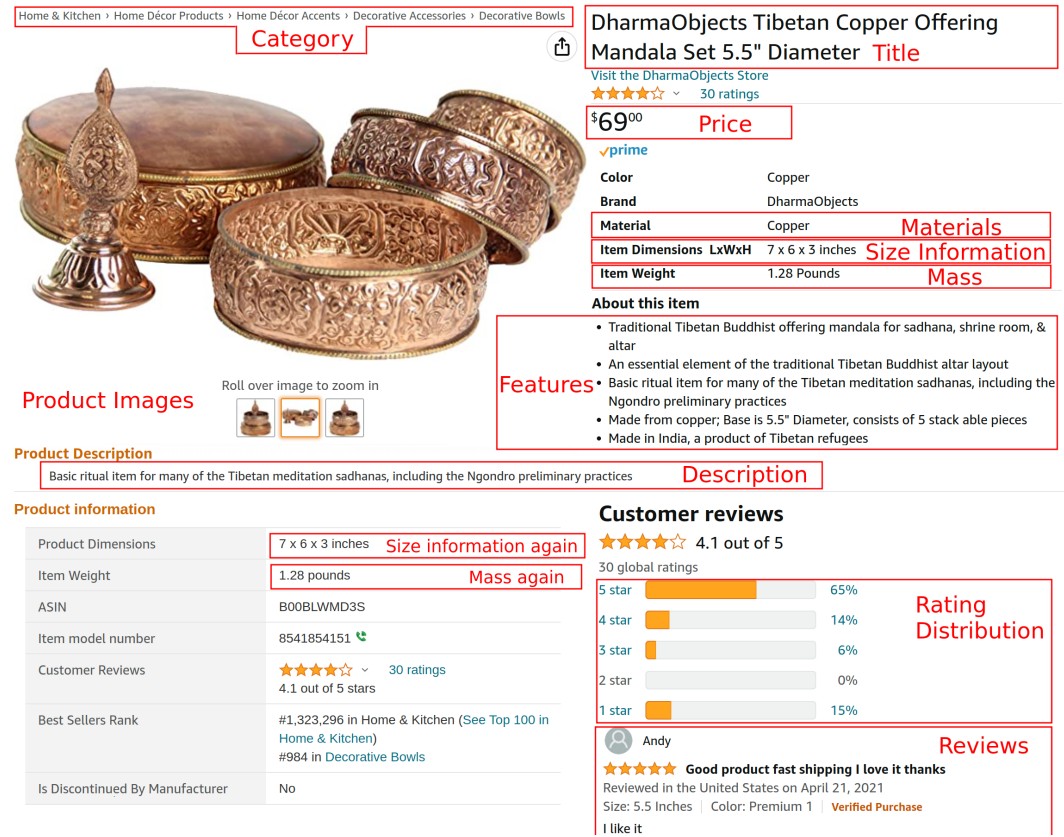

**Figure 9:** The anatomy of a product listing.

