# OpenReview forum: "An Extensible Multi-modal Multi-task Object Dataset with Materials"
_ICLR.cc/2023/Conference — ICLR 2023 poster_

### Official Review · Reviewer_NcVD · 2022-10-20

**Confidence:** 3
**Correctness:** 3
**Technical Novelty And Significance:** 2
**Empirical Novelty And Significance:** 3
**Recommendation:** 6

**Clarity, Quality, Novelty And Reproducibility:**

Clarity: clear and easy-to-follow. However, some details are missing, especially for the model illustration and experiments (hyperparameters, training, inference, etc.)

Quality: Well.

Novelty: a new and useful dataset that would be impactful for the community

Reproducibility: many details of the dataset construction process are given. However, more need to be confirmed and checked during use.

**Details Of Ethics Concerns:**

The images and text information may need to check to avoid possible ethics problems.

**Strength And Weaknesses:**

Pros:
+ An extensive large-scale dataset that would be useful for many directions and studies.

+ The data selection and annotation processes are described detailedly which would help the readers to use this dataset and enlarge it.

+ Considering the "materials that are not necessarily visible on the surface of an object, such as copper inside a wire", this is very interesting and would be inspiring for future works.

Cons:
- Lacking discussion about the object attribute datasets and methods (MIT-States, UT-Zap, etc.) though not all these works are related to products and are typically designed for common objects (but have some material attributes).

- Lacking discussion about compositional zero-shot learning (CZSL) which is very related to this work, especially for adding new attributes to objects (attribute-object compositions).

- It would be better to give a detailed bias analysis of the dataset itself and the annotation process. Because the authors use a lot of automatically labeling methods adopting DNN models. Moreover, the effect of longtail distribution is another important factor to analyze for both training and downstream applications.

- A list of data license and availability analysis?

- Though the proposed smart labeling system makes sense, the whole method designs are mainly common practices.

- It would be more solid to give a comprehensive analysis of the data quality verification as many labels are got not done manually (a clear ratio of pseudo labels).

- Lacking enough details of the models and experiments such as hyperparameters:

However, missing attributes can be accurately inferred from the available attributes: how accurate?

When precision and positive rate are sufficiently high: how much?

- "Even though Product Type and Category are both semantic classifications of the objects, they have surprisingly little overlap. Preliminary experiments show that models trained to predict one don’t benefit much from the inclusion of the other as an input": why?

- typo: we use uncertainty sampling(Lewis & Gale, 1994)： sampling (...)

- fig 7: explain what attribute is being labeled.

**Summary Of The Paper:**

This paper proposes a large-scale product dataset containing a lot of object properties such as listing text, mass, price, product ratings, and 182 material classes. The detailed collecting and implementation processes are given. Besides, a human-in-the-loop annotation system is proposed to actively add new attributes to the objects of the dataset. Given this labeling system, the authors show interesting additional annotation processes to add new attributes and verify the annotation quality quantitatively.

**Summary Of The Review:**

Overall, this work proposes a large-scale dataset that would be useful for future works. Besides, the labeling process makes sense and achieves reasonable quality as shown in the paper. However, a lot of details and explanations are missing to make this dataset paper more solid. I suggest a careful revision to improve the paper's quality. It is OK for me to accept this paper.

---

> ### Author Response · Authors · 2022-11-19
> **Response to Reviewer NcVD (part 3 of 3)**
>
> **Q8**: When precision and positive rate are sufficiently high: how much?
>
> This depends on the use case. We think the practitioner should decide. For example, when prototyping, the accuracy requirements are not as stringent as for production systems. And when human safety is involved, the requirements can be even more stringent. If many attributes are to be added, the trade-off between annotation cost and quality may dictate less accuracy.
>
> ---
>
> **Q9**: "Even though Product Type and Category are both semantic classifications of the objects, they have surprisingly little overlap. Preliminary experiments show that models trained to predict one don’t benefit much from the inclusion of the other as an input": why?
>
> High-level categories are easy enough to tell from images, text, etc., and Types don't have low-level distinctions, so they don't help much with either high- or low-level (in the taxonomy) category prediction. Furthermore, types are noisy. They mainly signify which template was used when creating the listing, but any listing can be made from any template. We have moved away from using Type in our models and have de-emphasized it in the updated paper.
>
> ---
>
> **Q10**:  typo: we use uncertainty sampling(Lewis & Gale, 1994)： sampling (...)
>
> Fixed. Thanks!
>
> ---
>
> **Q11**: fig 7: explain what attribute is being labeled.
>
> It's for the "is container" property. We have updated the text to reflect this.
>
> ---
>
> **Q12**:  The images and text information may need to check to avoid possible ethics problems.
>
> Thanks for pointing this out.  We agree on the importance of addressing these potential concerns. We have added an ethical considerations section to the end of the paper.
>
> Specifically, our actions to ensure no ethical concerns are three-fold. First, Amazon actively enforces its community standards on product listings and reviews. We therefore rely on this to ensure our dataset does not contain overt racism, foul language, nudity, illegal products, and harassment. Second, we recently employed Smart Labeling to mark images with human models or products of a sexual or religious nature, so they can be filtered when necessary. Third, we have consulted two experts on the ethics of AI datasets, both of whom have weighed in and concluded that these steps should be sufficient. We are always open to any additional suggestions or review steps, and thanks again for your comment.

---

> ### Author Response · Authors · 2022-11-19
> **Response to Reviewer NcVD (part 2 of 3)**
>
> **Q3**:  A list of data license and availability analysis?
>
> All data from Amazon is under copyright. Rights are typically owned by the original seller/photographer. However, as with most datasets, fair-use is permitted for academic purposes. Our curation and manual labels will be under a CC-BY-NC-4.0 license. The entire dataset will be made available on the project website upon publication.
>
> ---
>
> **Q4**:  Though the proposed smart labeling system makes sense, the whole method designs are mainly common practices.
>
> We are not aware of any systems that leverage a powerful object embedding like what we have created. We've also refined the active learning system significantly.
>
> We think that active learning is not being exploited as much as it should be in the deep-learning era. This is likely because efficiently training networks from scratch using active learning is an uncracked nut in the research field. We sidestep this issue by using classical learning on top of deep features. As far as we can tell, this approach is not a common one, though we're sure it has been done before. Any citations are welcome.
>
> ---
>
> **Q5**: It would be more solid to give a comprehensive analysis of the data quality verification as many labels are got not done manually (a clear ratio of pseudo labels).
>
> In section 3.2 we mention what proportion of the data contains a particular attribute. The synthetic attributes are the inverse of this proportion. So, for mass, $30.6$% of the labels are synthetic. For price, $28.4$% are synthetic. For materials, $63.4$% are synthetic. For Categories, $39.1$% are synthetic. And for ratings, $33.0$% are synthetic.
>
> ---
>
> **Q6**: Details of the models and experiments such as hyperparameters:
>
> We have added the following paragraph as well as two tables to the appendix under "Training Details":
>
> All models were trained using Pytorch and the AdamW optimizer. All models used a batch size of 512. All gradients were clipped to 32. Task losses were combined using a weighted average. The weights are in Table 4.  Text and vision models use an encoder-decoder architecture. For both, the encoder is frozen during the first epoch. Text and vision models were trained on 8xTitan RTX. Each took about three days to train. Everything models were trained on a 1xTitan RTX wokstation in about 12 hours. Each model type has its own learning rate schedule and weight-decay, which are listed in Table 5.
>
> Also, hyperparameters were added to the "Object Embedding" paragraph in the Appendix (page 15).
>
> ---
>
> **Q7**:  However, missing attributes can be accurately inferred from the available attributes: how accurate?
>
> In order to sanity check the accuracy of the synthetic category labels, we chose 100 random instances that were missing the category attribute. We manually examined the synthetic label assigned to each and found that 82 of those labels were either perfect or very good descriptions of the product.  Of the other 18, most weren't completely bad, there was just a better category for describing the object. We repeated this experiment on 100 objects with categories directly from Amazon, and found that 86 of the objects have good categories. However, we note that when synthetic labels fail, they are less likely to fail catastrophically.
>
> Since we use the Everything models to create the synthetic labels, **the performance of the Everything models in Table 1 represents expected accuracy of the synthetic labels.** We are happy with these metrics:
>
> * Price has a MnRE of $0.778$
>
> * Mass has a MnRE of $0.818$
>
> * Materials has a F1 of $91.6$
>
> Each level of the categories taxonomy matches ground truth $92.5$% of the time on average.
>
> Our predicted rating distribution achieves a KL-divergence of $0.320$ (Filled in rating information is a hypothetical answer to the question, "How would we expect this item to be rated if it were to get rated?")
>
> Furthermore, we show in Figure 4 that models trained with these predicted labels outperform models trained without the labels. We've added some discussion of this issue in the captions of Figure 4 and Table 1.

---

> ### Author Response · Authors · 2022-11-19
> **Response to Reviewer NcVD (part 1 of 3)**
>
> Thanks for the constructive comments and helpful feedback. We appreciate your recognition of the usefulness of the dataset for many directions and studies, and the uniqueness of our material taxonomy and approach to label materials that aren't surface visible. We have added additional experiments, diagrams, and textual details to address your questions and concerns. Changes to the paper are in blue.
>
> **Q1**:  Discussion about the object attribute datasets and methods (MIT-States, UT-Zap, etc.) though not all these works are related to products and are typically designed for common objects (but have some material attributes). Discussion about compositional zero-shot learning (CZSL) which is very related to this work, especially for adding new attributes to objects (attribute-object compositions).
>
> Thanks for the references, we've added them and some discussion as the last two paragraphs in Section 2. We have copied that discussion here:
>
> Attribute datasets:  The MIT-States and UT-Zappos (Isola et al., 2015; Yu & Grauman, 2014; 2017) datasets include object attributes, but are relatively small and focus on crowdsourcing the labels.
>
> Compositional Zero-shot Learning:  Like our work, CZSL (Misra et al., 2017; Nagarajan & Grauman, 2018; Purushwalkam et al., 2019; Li et al., 2020; Mancini et al., 2021) aims to provide attribute classifiers. The key challenge in these works, however, is that the training set is constructed to not contain object-attribute pairs that are present in the test set. For example, the test set would contain old bikes, but the training set only contains new bikes and old household items. On the other hand, we provide all the supervision we have to our models.
>
> ---
>
> **Q2**:  It would be better to give a detailed bias analysis of the dataset itself and the annotation process. Because the authors use a lot of automatically labeling methods adopting DNN models. Moreover, the effect of longtail distribution is another important factor to analyze for both training and downstream applications.
>
> For most of these attributes, Amazon is considered a source of ground truth. Rating and price information are considered to be fully accurate (though both often change), and there isn't a way for us to verify mass information without purchasing these objects, but we note that our mass classifiers trained on our dataset outperform the prior state-of-the-art, and do better when trained using our synthetic labels.
>
> Category information is chosen by sellers and they are incentivized to properly place their products. But to double check, we chose 100 objects with category labels from Amazon. We find that 86 of these categories make sense, though there can sometimes be other category labels that would also make sense. We repeated this experiment with objects from the training set that only have synthetic labels. In that case, 82 of the categories were good. This is not statistically significantly different, and we note that when synthetic categories fail, it's less likely to be a catastrophic failure.
>
> Material properties are also added by sellers and are not infallible. The majority of the effort we have spent on this project is on ensuring the quality of the materials labels. We have manually labeled the materials for 15% of the held-out set, and have used Smart Labeling for 16 material classes. We still see inaccuracy for some material types. We plan to produce a detailed analysis of the accuracy of the most common material properties before the dataset is released.
>
> Regarding the longtail distribution analysis, we would like to note that on our held-out sets, we achieve 91.6% accuracy on even the 6th (lowest) level of category prediction. This number is computed given the correct choices on the preceding 5 levels of the taxonomy.  This means that our classifier does a fairly good job with even the most fine-grained classifications which are in the long tail. If there are any other additional analyses that you believe we should include, we are definitely open to your suggestions.

---

### Official Review · Reviewer_8smg · 2022-10-25

**Confidence:** 2
**Correctness:** 3
**Technical Novelty And Significance:** 1
**Empirical Novelty And Significance:** 2
**Recommendation:** 6

**Clarity, Quality, Novelty And Reproducibility:**

The paper is clearly written. It proposes a new multi-task, multi-modal dataset dataset with rich annotations of attributes.

**Strength And Weaknesses:**

(+) The proposed dataset contains detailed object-centric rich annotations of attributes, including detailed material types, which seem not available in existing datasets.

(-) Some example use cases of this dataset could be introduced in the paper.

(-) Attributes can be independent from each other. In this case, the idea of predicting the missing attributes based on the existing attributes may not be reasonable.

(-) I am not sure if the proposed Smart Labeling pipeline will apply to other datasets for a bigger impact, because it seems some of its components are designed specifically for the EMMa dataset.

(-) Some details of the Smart Labeling pipeline are unclear to me. Why is the feature vector 4,542-dimensional though feature embeddings from image and text are both 768-dimensional?

**Summary Of The Paper:**

This work proposes a new multi-modal dataset, EMMa, which contains 2M+ Amazon product listings with rich annotations of attributes, including images, prices, materials, ratings, etc. In particular, it contains object-centric annotations for 182 material types. Finally, it proposes a Smart Labeling pipeline that allows users to add new annotations with reduced effort.

**Summary Of The Review:**

In summary, I think this is a valuable dataset with rich annotations of attributes, which can be potentially used for multi-modal or multi-task research. But I am not sure about the potential use cases and their impacts for now.

---

> ### Author Response · Authors · 2022-11-19
> **Response to Reviewer 8smg**
>
> Thanks for the constructive comments and helpful feedback. We appreciate your recognition of the effort we made to collect this dataset, and our detailed object-centric rich annotations and attributes. In order to address your concerns, we have run a new experiment, created a new diagram, and re-written portions of the paper. Changes in blue.
>
> **Q1**: Some example use cases of this dataset could be introduced in the paper.
> In the third paragraph of the Introduction, we mention that models trained on this data could be useful for roboticists, recycling facilities, consumers, marketers, retailers, and product developers. Some specific ideas are listed below.
>
> * Roboticists can use the images to help inform how they should manipulate objects, which objects are good to complete which tasks, and which objects should be avoided.
>
> * The materials attribute could be invaluable to recycling centers, which often strive to automate mixed-stream sorting. The way we see this working is that they can use an image embedding to help them train their models on their existing datasets.
>
> * Consumers could perhaps install an extension that helps them find products that satisfy their criteria. They could also use the model to determine if a price is too good to be true or is not a good deal.
>
> * Marketers could use the models to help them choose prices for a product and to advertise which features of a product are unique within the product's category. They could also use the models to learn how to take better pictures or write better product descriptions.
>
> * Retailers can use the mass attribute to help with logistics, or the ratings attribute to determine which products to stock.
>
> * Product developers could use the dataset to figure out what materials an object should be made out of to maximize profit or to maximize buyer satisfaction.
>
> * Furthermore (as we mention in the fourth paragraph,) we believe that this could be a great multi-task benchmark for both computer vision and NLP.
>
> We don't believe this list is exhaustive, but we do think that there are many uses. One of the main contributions to this work, Smart Labeling, seeks to allow practitioners to come up with their own use cases that we could never foresee.
>
> ---
> **Q2**:  Attributes can be independent from each other. In this case, the idea of predicting the missing attributes based on the existing attributes may not be reasonable.
>
> In order to sanity check the accuracy of the synthetic category labels, we chose 100 random instances that were missing the category attribute. We manually examined the synthetic label assigned to each and found that 82 of those labels were either perfect or very good descriptions of the product.  Of the other 18, most weren't completely bad, there was just a better category for describing the object. We repeated this experiment on 100 objects with categories directly from Amazon, and found that 86 of the objects have good categories. However, we note that when synthetic labels fail, they are less likely to fail catastrophically.
>
> Since we use the Everything models to create the synthetic labels, **the performance of the Everything models in Table 1 represents expected accuracy of the synthetic labels.** We are happy with these metrics.
>
> Furthermore, we show in Figure 4 that models trained with these predicted labels outperform models trained without the labels. We've added some discussion of this issue in the captions of Figure 4 and Table 1.
>
> Still, it is strictly true that some of the predicted labels may not be accurate. However, we aren't just using the other attributes, but also the text and the images, so we are bringing a lot of information to bear.
>
> ---
>
> **Q3**:  I am not sure if the proposed Smart Labeling pipeline will apply to other datasets for a bigger impact, because it seems some of its components are designed specifically for the EMMa dataset.
>
> It's true that the Smart Labeling pipeline was designed to work specifically with EMMa. Just as much, the dataset was designed to be good for Smart Labeling. However, we briefly discuss how Smart Labeling can be useful for dataset filtering for more typical datasets in the second-to-last paragraph of Section 5.
>
> ---
>
> **Q4**:  Some details of the Smart Labeling pipeline are unclear to me. Why is the feature vector 4,542-dimensional though feature embeddings from image and text are both 768-dimensional?
>
> To make this clearer, we've re-written the Object Embedding paragraph from Section 5, and added a new diagram to the Appendix. The input to the bottleneck model is more than those two vectors.

---

> > ### Comment · Reviewer_8smg · 2022-11-28
> > **Thank you for the response**
> >
> > Thank you for the response. It helped me understand the possible use cases.

---

### Official Review · Reviewer_AKRZ · 2022-10-25

**Confidence:** 3
**Correctness:** 3
**Technical Novelty And Significance:** 3
**Empirical Novelty And Significance:** 3
**Recommendation:** 6

**Clarity, Quality, Novelty And Reproducibility:**

The presentation of the dataset and method are quite clear. However the motivation for is missing for some parts and the discussion for others. The dataset seems to be novel in its multi-modality and multi-purpose applications, the size of the validation and test split seem a bit small though.

**Strength And Weaknesses:**

Strength:
The dataset contains more than 2m objects collected over several month.
The proposed interactive labeling strategy is simple and intuitive and can be used to add various new labels creating a continously growing database for many tasks.

Weaknesses:
It would be interesting to split the contributions/workload of the different steps word search, active learning, correction and review in a sperate ablation study to better understand the impact of each step.

**Summary Of The Paper:**

The paper presents a novel multi-modal dataset based on Amazon items, with image, free text and structured fields as input or output.
They further propose a novel material taxonomy and propose an interactive labeling process to allow the community to add new labels to the dataset.

**Summary Of The Review:**

In total the novel dataset represents a large database of object centric and related image, text and categorical data.
The idea of publically extending the labels is also interesting, however the question of quality control, also for the imputation conducted by the authros themselves, remains open
After reading the rebuttal I better understand the semi-automatic labeling procedure and evaluation.
While I am still interested in the ablation of the 3 steps (word search, active learning, correction), I think the dataset is a fruitful contributon to the community.

---

> ### Author Response · Authors · 2022-11-19
> **Response to Reviewer AKRZ (part 3 of 3)**
>
> **Q9**: It would be interesting to split the contributions/workload of the different steps word search, active learning, correction and review in a separate ablation study to better understand the impact of each step.
>
> We commit to having at least one property ablated like this in the final paper, but because each stage can be visited and revisited, this type of ablation requires manually re-annotating the dataset from scratch for each variable to ablate. This is because the results depend on choices made by the annotator. We can't merely include only data or features added in each step because the active learning process decides which data to label based on what has already been labeled and which text features already have been added.
>
> We think that active learning mode increases accuracy the fastest initially, but eventually becomes saturated. When it becomes saturated, correction mode takes over as the most important. Word search mode is only necessary to kick off learning or if the model neglects important classes of objects. Features can help a bit, but there are only so many that one can ever think to add. Review mode is absolutely necessary for good performance because in our experience manual annotation has >2% error rate.
>
> ---
>
> **Q10**:  Finally the limitations of the method should be discussed e.g that the logistic regression is limited by the pre-trained image and text features.
>
> We have added a discussion on limitations to Section 5. **In the original draft, we provided a strategy to overcome the embedding's limitation somewhat in the "features" paragraph at the bottom of page 7.** Adding manual features (often just simple word-matching features) can be very useful when the object embedding is insufficient. Also, it's not just text and image features. The embedding includes information from the rest of the attributes as well.

---

> > ### Comment · Reviewer_AKRZ · 2022-12-10
> > **Rebuttal review**
> >
> > Thank you for the detailed reponse. Most of my questions were answered and I updated my intial review.

---

> ### Author Response · Authors · 2022-11-19
> **Response to Reviewer AKRZ (part 2 of 3)**
>
> **Q5**:  As noted by the authors the original dataset was incomplete and has been (semi)automatically completed. This leads to the question of label quality for future tasks and whether the imputated labels were somehow annotated.
>
> We looked at the inferred categories for 100 random objects that did not have categories in the source data. 82 of these objects had inferred categories that were perfect or very good for describing the object. Of the other 18, most weren't completely bad, there was just a better category for describing the object. We repeated this experiment on 100 objects with categories directly from Amazon, and found that 86 of the objects have good categories. However, we note that when synthetic labels fail, they are less likely to fail catastrophically.
>
> Since we use the Everything models to create the synthetic labels, **the performance of the Everything models in Table 1 represents expected accuracy of the synthetic labels.**
>
> For all tasks, we note that models trained on these automatic labels do better on the test sets as shown in Figure 4. Another reason it's a good idea to fill in these labels is that the object embeddings from section 5 can be trained in a way that assumes all data is present. We have updated the text to make it clearer in the paper.
>
> ---
>
> **Q6**:  The authors also mention multiple times the difficulty of communicating how data should be labeled for human annotators, however they do not discuss how their proposed process improves this issue.
>
> The solution involves the practitioner themselves manually labeling a small amount of data that can be used to train a classifier to label the entire dataset. Communicating with human annotators is no longer necessary because the practitioner only needs to label a small amount of data themselves. There no longer are any human annotators apart from the practitioner. We've further clarified this in the "Smart Labeling Advantages" paragraph in section 5.
>
> ---
>
> **Q7**:  It would be interesting to discuss the design choice for the embedding(s) especially for text and image. An ablation study might help to show the amount of loss during the autoencoding.
>
> The embedding described in Section 5 is a holistic object embedding that includes information from not only text and images but also materials, prices, masses, categories, etc. The auto-encoder has a very small standardized L2 loss of $0.0054$, which implies that each input feature is reconstructed to be within $7.3$% of its variance on average. Furthermore, using this embedding, we can reconstruct the category with $99.8$% accuracy, the materials with an F1 of $98.45$, the mass with a Pearson's $R^2$ of $0.998$ correlation with the ground truth, and the price with a Pearson's $R^2$ of $0.960$.
> Here are L2's for various embedding sizes:
> |         2048                   |    1536      |   1024         |      512     |
> |------------------------------|---------------|-----------------|--------------|
> |**$0.0054$**  | $0.0154$ |  $0.04951$  | $0.2212$ |
>
> We have switched to using an embedding of size 2048. We have also added more details to the embedding section in the paper, including a new diagram in the Appendix (Figure 8, page 16).
>
> ---
>
> **Q8**: While the labeling method has a simple design and is explained clearly. The authors do not discuss the labeling effort needed in the different stages.
>
> Figure 5 in the original paper gives a rough expectation of how much time should be taken on each stage. Features mode typically takes minutes in total. It only takes seconds to add a text-matching feature, and that can be done at any time during the annotation process. Usually we have about a dozen or so text matching features. Word search mode typically also takes only minutes. It's mainly important for seeding a small initial set of positives and negatives to kick-off the learning. Active Learning mode takes the most time. Practitioners should plan on spending two or more hours depending on how easily positives and negatives can be identified. Correction mode is a precision and recall boosting step and takes up to an hour depending on how accurate the model already is. Finally, review mode typically only requires a few minutes depending on how many annotation errors were made on other stages of the pipeline.

---

> ### Author Response · Authors · 2022-11-19
> **Response to Reviewer AKRZ (part 1 of 3)**
>
> Thank you for your insightful comments. They have definitely helped to make the paper better. We really appreciate that you like our novel material taxonomy, and the simple and intuitive interactive labeling strategy, which enables the continuous growing of the dataset for many more tasks. In order to address your concerns, we added new data to the train, validation, and test sets, ran several new experiments, and re-wrote portions of the paper.  Changes in the updated paper are in blue.
>
> **Q1**: How many new items are added by the fusion of the initial 2m images with image2mass and UCSD product rating? What is the benefit?
>
> We've included a more detailed and up-to-date discussion on the dataset merging and filtering process in the updated paper. We've highlighted updates in Section 3.1 and in the Appendix under "Filtering and Data Cleaning".  The benefit of merging entries with the same ASIN from multiple datasets is that we are able to get objects annotated with attributes that aren't present in the original datasets. For example, an object may be labeled with mass in the image2mass dataset only and categories in the UCSD dataset only. By merging, that object now has both.
>
> ---
>
> **Q2**: How was the number of validation and test instances chosen? Number of validation and test instances comparably low compared to number of product categories (11,592 or 671 with > 1000 instances).
>
> We initially chose a relatively small held-out set so that we could manually annotate the materials for a significant number of them in a reasonable amount of time. We manually corrected the materials for 15% of the held-out set, focusing on instances with the highest losses.
>
> We have added 800k instances to the dataset, and **more than doubled the train/validation splits to contain 56k objects with 280k images total.** Each held-out instance has more than the average number of images. All instances in the held-out sets have all attributes present (not synthetic) and there just aren't that many more instances like this.
>
> It's true that some categories are still not well represented in the held-out sets. This is inevitable because some of these are rare on Amazon. But since this is taxonomic classification, we can check each instance's classification on every node on the path down to the leaf class (the leaf typically being the rare class). So we can still get an accurate read on how well an average instance is classified.
>
> ---
>
> **Q3**: Some classes of the material taxonomy are more texture or surface properties rather than materials e.g. microfibre, plush and fleece and a bit of discussion on the idea behind the taxonomy would be interesting.
>
> The taxonomy was designed in a data-driven way. Many entries in the raw datasets contained these types of "materials". For example, 'plush' was present in thousands of products' materials entries. We decided to include materials like these for a number of reasons.
>
> First, many of these imply their parents, i.e., microfiber implies fabric.
>
> Second, philosophically, most materials we think about are not pure elements, compounds or mixtures. Leather and wood, for example, are about the configuration of molecules, not the base molecules themselves.
>
> Finally, we can envision uses for knowing about these types of "materials". For example the weave of fabric. Some weaves are denser or more expensive than others, so that has implications for price and mass, and may even be a useful signal for custom features.
>
> We have added this discussion to the Appendix and pointers in the caption of Figure 2.
>
> ---
>
> **Q4**:  The evaluation of the models trained with the text and image embeddings is missing the descriptions/references for the metrics e.g. minimum ratio error.
>
> The paper has been updated to include more discussion on metrics and losses in the Appendix (See subsection 4). We also have the citation for MnRE more consistently in the main text.
>
> MnRE (Min Ratio Error) is defined in Standley et al. (2017). We find that this metric is well-suited to our needs and it also allows us to compare to the 2017 paper. A value of 1.0 is a perfect prediction, and a value of 0.0 is as bad as predictions can get.

---

### Decision · Program_Chairs · 2023-01-20

**Decision:**

Accept: poster

**Justification For Why Not Higher Score:**

Yet another dataset paper, with innovation related to labelling process

**Justification For Why Not Lower Score:**

Useful to let ICLR researchers know about large multi-modal product listing dataset

**Metareview: Summary, Strengths And Weaknesses:**

The authors propose a new multi-modal dataset with 2M+ Amazon product listings with rich annotations of attributes, including images, prices, materials, ratings, etc. They propose an interactive labelling process to actively add new attributes to the objects of the dataset and verify the annotation quality quantitatively. This can be useful as it allows practitioners to efficiently add new tasks and object attributes.


**Note From Pc:**

if the above contains the word "oral" or "spotlight" please see: "oral" presentation means -> notable-top-5% and "spotlight" means -> notable-top-25%. As stated in our emails, we are disassociating presentation type from AC recommendations

**Summary Of Ac-Reviewer Meeting:**

NA as all 3 reviewers score 6